# Extended Lymph Node Dissection May Not Provide a Therapeutic Benefit in Patients with Intermediate-to High-Risk Prostate Cancer Treated with Robotic-Assisted Radical Prostatectomy

**DOI:** 10.3390/cancers17040655

**Published:** 2025-02-14

**Authors:** Noriyoshi Miura, Masaki Shimbo, Dai Okawa, Miki Sakamoto, Naoya Sugihara, Takatora Sawada, Shunsuke Haga, Haruna Arai, Keigo Nishida, Osuke Arai, Tomoya Onishi, Ryuta Watanabe, Kenichi Nishimura, Tetsuya Fukumoto, Yuki Miyauchi, Tadahiko Kikugawa, Takato Nishino, Fumiyasu Endo, Kazunori Hattori, Takashi Saika

**Affiliations:** 1Department of Urology, Ehime University Graduate School of Medicine, Toon 791-0295, Japan; dai.b.m.0816@gmail.com (D.O.); mikinako0318@gmail.com (M.S.); ns.fish708majt@gmail.com (N.S.); takapowa999@gmail.com (T.S.); samourai_light_edt@yahoo.co.jp (S.H.); susiehanji02i@yahoo.co.jp (H.A.); keigo9027@yahoo.co.jp (K.N.); kin.niku.pain@gmail.com (O.A.); onishi.tomoya.ey@ehime-u.ac.jp (T.O.); watanabe.ryuta.cu@ehime-u.ac.jp (R.W.); i401063@hotmail.co.jp (K.N.); siawase110bann@yahoo.co.jp (T.F.); uroyuukictb12@yahoo.co.jp (Y.M.); takikuga@m.ehime-u.ac.jp (T.K.); saika.takashi.ol@ehime-u.ac.jp (T.S.); 2Department of Urology, St. Luke’s International Hospital, Tokyo 104-8560, Japan; mashimbo@luke.ac.jp (M.S.); nisitaka@luke.ac.jp (T.N.); endofum@luke.ac.jp (F.E.); kazhat@luke.ac.jp (K.H.)

**Keywords:** prostate cancer, extended lymph node dissection, robotic-assisted radical prostatectomy

## Abstract

This study evaluated the therapeutic significance of extended pelvic lymph node dissection (ePLND) in patients with intermediate- and high-risk prostate cancer undergoing robotic-assisted radical prostatectomy (RARP). Current guidelines provide conflicting recommendations on the efficacy of ePLND, with some suggesting its potential in improving oncological outcomes like biochemical recurrence-free survival (bRFS), while others highlight the associated risks of longer surgical time and complications. By using a propensity-matched analysis across two institutions with differing ePLND policies, this study aimed to clarify whether ePLND impacts bRFS or merely adds operative risks. The findings may help to refine clinical decision-making, emphasizing the importance of patient selection and balancing oncological benefits with potential harm. This research contributes to ongoing discussions within the urological community and may guide future studies and policy updates on the role of ePLND in prostate cancer management.

## 1. Introduction

Various guidelines recommend robot-assisted radical prostatectomy (RARP) as a treatment option for localized or locally advanced prostate cancer (PCa) [1,2,3]. Pelvic lymph node dissection (PLND) is also recommended as a treatment for patients with intermediate- or high-risk prostate cancer [4]. PLND plays an important role in accurate PCa staging and may have potential benefits in micrometastasis removal [5,6]. The European Association of Urology (EAU) guidelines call for extended pelvic lymph node dissection (ePLND) in patients with a ≥5% probability of lymph node metastasis based on the Briganti nomogram, or in patients with high-risk PCa [7]. Similarly, the National Comprehensive Cancer Network (NCCN) guidelines recommend ePLND for patients with intermediate- and high-risk PCa. However, the American Urological Association (AUA) guidelines state that while PLND is the most effective procedure for detecting lymph node metastasis, evidence is lacking regarding its therapeutic efficacy [3].

While some studies have shown therapeutic advantages, including lower biochemical recurrence-free survival (bRFS) and cancer-specific mortality (CSM), in patients treated with more extensive PLND during radical prostatectomy (RP) [8,9,10,11], recent studies have shown that the presence or degree of PLND has no statistical impact on oncologic outcomes such as bRFS [4,12,13]. Moreover, ePLND is significantly associated with longer operative time and increased perioperative and postoperative complications, including lymphadenopathy, bleeding, and thromboembolic events [14,15,16,17].

Due to the lack of prospective randomized trials, the therapeutic advantage of PLND remains controversial. Therefore, this study evaluated whether PLND improves BCR in patients with PCa undergoing RARP, using a propensity-matching method with cases from facilities with equivalent surgical techniques and hospital size to minimize bias.

## 2. Materials and Methods

### 2.1. Study Population

Between July 2012 and November 2022, 1879 patients with PCa underwent RARP at two large Japanese institutions, namely, Ehime University and St. Luke’s International Hospital. All patient data were retrospectively collected in an electronic database and analyzed after obtaining Institutional Review Board approval (IRB no. 22100007). Only patients with NCCN moderate- or high-risk localized prostate cancer were included in the analysis. All patients were provided with the available information on clinical tumor stage, biopsy Gleason score (GS), and prostate-specific antigen (PSA) status. Patients with unknown PLND results, pathologic tumor stage, or operative time, as well as those who received neoadjuvant or adjuvant therapy, were also excluded from the study. After applying these selection criteria, the final analysis encompassed 1002 patients.

The two study facilities treat similar numbers of patients, perform similar numbers of surgeries, and utilize similar surgical techniques. However, their indications for PLND differ: Ehime University performs non-PLND for patients with intermediate-to-high-risk PCa, while St. Luke’s Hospital performs ePLND.

### 2.2. Surgical Approach

All patients underwent RARP. ePLND included the internal iliac, obturator, and common iliac lymph nodes. All patients in whom PLND was performed underwent extended PLND, as previously described [9,18].

### 2.3. Endpoints

The primary endpoint was the therapeutic effect of ePLND on bRFS. The secondary endpoints were the differences in surgical outcomes and complications between the groups, and the benefit of ePLND in terms of nodal staging. BCR was defined as two consecutive PSA values of ≥0.2 ng/mL. Surgical complications were evaluated using the Clavien–Dindo classification [19].

### 2.4. Statistical Analysis

All statistical analyses were performed using EZR [20], a modified version of R commander that adds statistical functions and is frequently used in biostatistics.

Categorical variables were compared using Fisher’s exact or χ^2^ tests. Quantitative variables are expressed as medians and interquartile ranges. Student’s *t*-test or the Mann–Whitney U test was used to compare group differences. Qualitative variables were compared using a chi-squared test. The Kaplan–Meier method was used to compare bRFS between the ePLND and non-PLND groups. Hazard ratios (HRs) and 95% confidence intervals (CIs) were obtained using a multivariate Cox regression analysis in a propensity-adjusted model. We performed a propensity-matching analysis using the nearest neighbor matching method with a 1:1 matching ratio. The caliper was set to 0.2 standard deviations of the propensity score (Figure 1). After creating a pseudo-population, background imbalances between groups were removed using a propensity score-based method to estimate the mean treatment effect without bias. Logistic regression analysis was used to calculate propensity scores using parameters such as initial PSA, biopsy GS, and clinical tumor stage. P-values of < 0.05 were considered statistically significant.

## 3. Results

### 3.1. Baseline Characteristics

Between July 2012 and November 2022, 1879 patients underwent RARP at the two study institutes. Among these, patients treated with neoadjuvant therapy (*n* = 47) or with limited PLND or unclear lymph node dissection area (*n* = 285) were excluded. All patients in the study were diagnosed by MRI before surgery. Finally, we analyzed 632 patients with intermediate-risk and 370 patients with high-risk PCa, including 541 who underwent ePLND and 461 who did not undergo PLND.

The characteristics of the original cohort and propensity-score-matched cohort are shown in Table 1. In the original cohort, the preoperative clinicopathological features differed significantly between the two groups. Patients in the ePLND group were younger (*p* < 0.001) and had significantly higher clinical stage PCa (*p* < 0.001) biopsy GS (*p* < 0.001) with intermediate risk. Among the high-risk patients, patients undergoing ePLND were significantly younger (*p* < 0.001) and had a lower initial PSA status (*p* = 0.01) and biopsy GS (*p* < 0.001). Propensity matching resulted in a comparison of 221 cases, each with intermediate risk, and 124 cases, each with high risk. In the matched cohorts, except for age, the preoperative clinicopathological variables did not differ significantly between the ePLND and non-PLND groups.

In the ePLND group, a median of 18 nodes were dissected (interquartile range: 14–25). Lymph node metastases were detected in 12 (3.8%) and 22 (9.8%) patients in the intermediate- and high-risk groups, respectively. The patients in the intermediate group each showed a single positive lymph node. In the high-risk group, fifteen patients had one positive lymph node, four patients had two positive lymph nodes, and two patients had three or more positive lymph nodes.

### 3.2. Perioperative Clinical Characteristics and Pathological Outcomes (Propensity-Score-Matched Cohort)

Compared with the ePLND group, the non-PLND group had significantly shorter median operative times (intermediate-risk group: 175 vs. 325 min, *p* < 0.001; high-risk group: 180 vs. 327 min, *p* < 0.001) and lower median blood loss (intermediate-risk group: 0 vs. 150 mL, *p* < 0.001; high-risk group: 0 vs. 150 mL, *p* < 0.001) (Table 2).

The incidence of surgical complications was significantly higher in the ePLND group than in the non-ePLND group (all grades: 159 [29.4%] vs. 37 [8.0%], *p* < 0.0001; grade ≥ 3: 33 [6.1%] vs. 10 [2.2%], *p* < 0.0001). The most frequent complications related to ePLND were lower extremity edema (9.8%), pelvic hematoma (1.7%), and neuropathy (1.3%) (Table 3). Of the 53 cases of lower limb edema, 6 cases (11.3%) required the long-term use of stockings. Intra-pelvic hematoma was observed in eight cases. Seven cases improved with conservative treatment, and one case underwent drainage. Peripheral sensory neuropathy was observed in seven cases, but all cases improved spontaneously.

### 3.3. Oncological Outcomes

The median follow-up period was significantly longer in the ePLND group than in the non-PLND group (77 vs. 34 months, *p* < 0.001). After 1:1 propensity score matching, the 3-year bRFS rates did not differ significantly between ePLND and non-PLND groups in intermediate-risk patients (92.4% vs. 91.6%, *p* = 0.821) and high-risk patients (77.8% vs. 81.3%; *p* = 0.947) (Figure 2).

Propensity-adjusted multivariate Cox regression analysis identified initial PSA, pathological tumor stage (high risk only), and positive surgical margins as independent prognostic factors for bRFS, but not ePLND (Table 4).

## 4. Discussion

In the present study, we examined 1002 patients with intermediate- or high-risk PCa who underwent RARP. Overall, our results using the propensity-score-matched analysis showed no significant differences in oncological outcomes between patients with NCCN IR or HR PCa regardless of ePLND at RARP. These findings suggest that omitting PLND in RARP for the treatment of IR or HR PCa may not affect oncological outcomes. The median number of lymph node dissections in this study was 18. This is comparable to the median numbers in expanded RCTs, namely 14 [21] and 17 [13]. A 2017 systematic review and meta-analysis reported no significant differences in oncological survival between men who did and did not undergo ePLND [5]. Two RCTs reported that ePLND improved N-staging compared with limited PLND but did not improve bRFS using the extended template [12]. Future trials to compare extended to no-node dissection in PCa have been suggested [21]. Recently, Touijer et al. updated the results of a randomized trial with 4.2 years of follow-up, reporting consistently similar bRFS rates between patients who underwent standard treatment and patients who underwent ePLND [12].

Mandel et al. reported a low detection rate of lymph node metastases in intermediate-risk patients with GSs of ≤6, cT 2b or less, and PSA 10–20 ng/mL. Moreover, the BCR did not differ significantly according to lymph node dissection, suggesting that lymph node dissection may be unnecessary not only in low-risk cases but also in some intermediate-risk cases [22].

Additionally, one study reported that bRFS did not differ between PLND and no PLND, even in high-risk cases [23]. However, the extent of lymph node dissection was not determined in the multicenter study and the number of lymph nodes dissected was small. In contrast, the results of the present study demonstrated no significant differences in bRFS between ePLND and non-PLND patients with a mean of 18 dissected lymph nodes, a number comparable to those in previous reports [14,21].

Systematic reviews have shown that the detection rate of positive lymph nodes increases with the number of lymph nodes removed by extended lymph node dissection [17,24,25]. One of the most significant aspects of diagnostic lymph node dissection is its impact on subsequent treatment strategies. The EAU guidelines recommend observation without adjuvant therapy in patients with one or fewer positive lymph nodes [1]. In this study, only six cases (1.3%) in the high-risk group and no cases in the intermediate-risk group had two or more lymph node metastases. Thus, the number of cases that would affect the postoperative treatment plan was extremely small. Cacccoamani et al. systematically evaluated the impact of the extent of ePLND on perioperative morbidity in patients undergoing RP. Overall, they observed one or more postoperative complications in 10,401 (14.1%) patients. The integrated meta-analysis revealed that ePLND was associated with a significantly increased risk of all intraoperative and postoperative complications compared with limited PLND. In particular, lymphocele formation and thromboembolic events were strongly associated with ePLND [14]. Furthermore, it has been reported that the frequency of complications in frail patients exceeds 50% [26], and in cases where the risk of complications due to surgical invasion or the risk of disadvantages due to prolonged surgery time is high, omitting lymph node dissection might be considered.

This study has some limitations. First, this was a retrospective study, and a selection bias toward patients who underwent RARP with PNLD was present. We sought to minimize this bias as much as possible by comparing facilities of similar size and number of surgical cases and by eliminating background differences through propensity matching. Regarding the equivalence of surgical techniques at the two facilities, RARP+ePLND was performed on patients with very high-risk PCa at both facilities, and the surgical outcomes were equivalent. Second, RP specimens were evaluated by pathologists at each facility, with no central pathologic review. Third, the median follow-up period in this study was short (31.9 months) and only bRFS was evaluated, with no assessment of metastasis-free survival (MFS) or overall survival. While long-term evaluation is important, this study considered bRFS as the endpoint owing to a report suggesting that PSA is a useful biomarker for predicting MFS in patients with PCa after RP [27]. Moreover, multiple randomized clinical trials have shown that MFS is a reliable surrogate endpoint for overall survival [28,29]. Given the limited number of long-term follow-up cases in the ePLND(-) group, further studies with extended follow-up are warranted.

## 5. Conclusions

These results suggest that ePLND may not be necessary in intermediate- to high-risk PCa patients undergoing RARP, although further research with longer follow-up periods is required. Thus, future randomized controlled trials evaluating the therapeutic significance of ePLND to clarify patient selection and the extent of dissection are warranted.

## Figures and Tables

**Figure 1 cancers-17-00655-f001:**
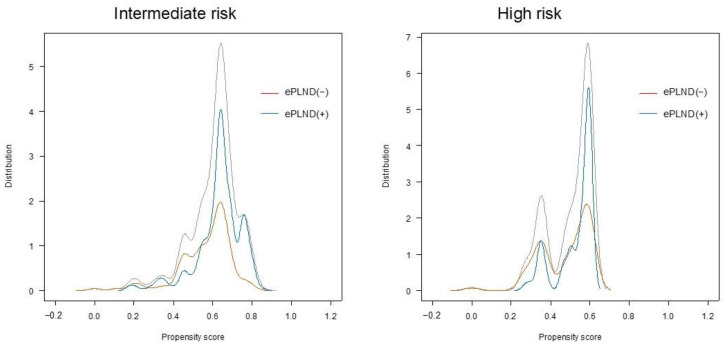
Distribution of propensity scores.

**Figure 2 cancers-17-00655-f002:**
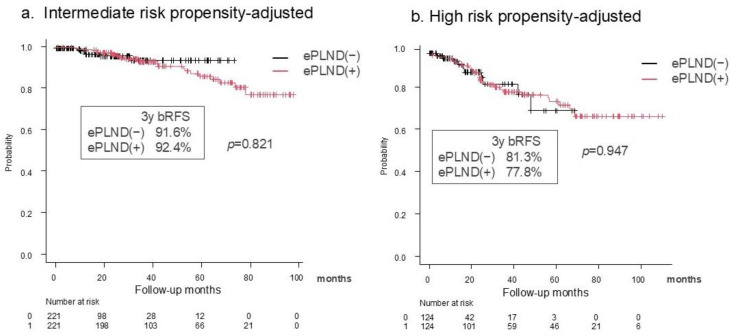
Kaplan–Meier curves for BCR-free survival with and without lymph node dissection in the propensity-score-matched cohort of intermediate- (**a**) and high-risk (**b**) patients, according to the NCCN.

**Table 1 cancers-17-00655-t001:** Preoperative clinicopathological features of patients, stratified by the ePLND (Upper: Original cohort, Lower: Propensity-score-matched cohort).

Variable (Original Cohort)	ePLND (−) *(n* = 315)	ePLND (*n* = 317)	*p* Value	ePLND (–) (*n* = 146)	ePLND (*n* = 224)	*p* Value
Age, median (IQR) yr	71 (67–83)	67 (62–79)	<0.001	71 (66–76)	68 (62–73)	<0.001
Preoperative PSA (ng/mL)						
<10	232 (73.7%)	254 (80.1%)	0.0592	91 (70.1%)	165 (77.4%)	0.0101
10≤□<20	83 (26.3%)	63 (19.9%)		37 (26.0%)	49 (20.1%)	
20≤				18 (3.9%)	10 (1.8%)	
Prostate biopsy ISUP Gleason Grading						

1	38 (12.1%)	3 (0.9%)	<0.001	6 (4.1%)	3 (1.3%)	<0.001
2	156 (49.5%)	220 (69.4%)		12 (8.2%)	18 (8.0%)	
3	121 (29.7%)	94 (29.7%)		15 (10.3%)	10 (4.5%)	
4	0 (0%)	0 (0%)		94 (64.4%)	125 (55.8%)	
5	0 (0%)	0 (0%)		19 (13.0%)	68 (30.4%)	
cT stage T1a-c	88 (28.0%)	42 (13.2%)	<0.001	27 (18.5%)	24 (10.7%)	0.234
T2a	143 (45.4%)	195 (61.5%)		70 (47.9%)	120 (53.6%)	
T2b	24 (7.6%)	13 (4.1%)		11 (7.5%)	15 (6.7%)	
T2c	60 (19.0%)	67 (21.1%)		23 (15.8%)	45 (20.1%)	
T3a	0 (0%)	0 (0%)		15 (10.3%)	20 (8.9%)	
T3b	0 (0%)	0 (0%)		0 (0%)	0 (0%)	
Variable (Propensity-score-matched cohort)	ePLND (–) (*n* = 221)	ePLND (*n* = 221)	*p* value	ePLND (–) (*n* = 124)	ePLND (*n* = 124)	*p* value
Age, median(IQR) yr	71 (66–74)	67 (62–71)	<0.001	71 (65–76)	67 (62–73)	<0.001
Preoperative PSA (ng/mL)						
<10	178 (80.5%)	180 (81.4%)	0.904	81 (65.3%)	87 (70.2%)	0.733
10≤□<20	43 (19.5%)	41 (18.6%)		34 (27.4%)	30 (24.2%)	
20≤				9 (7.3%)	7 (5.6%)	
Prostate biopsy ISUP Gleason Grading						

1	4 (1.8%)	3 (1.4%)	1	3 (2.4%)	2 (1.6%)	0.930
2	138 (62.4%)	138 (62.4%)		11 (8.9%)	14 (11.3%)	
3	79 (35.7%)	80 (36.2%)		9 (7.3%)	7 (5.6%)	
4	0 (0%)	0 (0%)		82 (66.1%)	83 (66.9%)	
5	0 (0%)	0 (0%)		19 (15.3%)	18 (14.5%)	
cT stage T1a–c	39 (17.6%)	38 (17.2%)	0.753	16 (12.9%)	18 (14.5%)	0.995
T2a	133 (60.2%)	125 (56.6%)		65 (52.4%)	61 (49.2%)	
T2b	9 (4.1%)	9 (4.1%)		9 (7.3%)	9 (7.3%)	
T2c	40 (18.1%)	49 (22.2%)		20 (16.1%)	21 (16.9%)	
T3a	0 (0%)	0(0%)		14 (11.3%)	15 (12.1%)	
T3b	0 (0%)	0(0%)		0 (0%)	0 (0%)	

**Table 2 cancers-17-00655-t002:** Perioperative clinical characteristics and pathological outcomes stratified by the ePLND. (Propensity-score-matched cohort).

Variable	ePLND (–) (*n* = 221)	ePLND (*n* = 221)	*p* Value	ePLND (–) (*n* = 124)	ePLND (*n* = 124)	*p* Value
Operation time median(IQR) min	175 (150–207)	325 (299–355)	<0.001	180 (153–217)	327 (293–354)	<0.001
Console time median(IQR) min	126 (99–149)	267 (240–292)	<0.001	125 (101–161)	271 (249–302)	<0.001
Bleeding median(IQR) mL	0 (0–100)	150 (100–250)	<0.001	0 (0–100)	150 (100–250)	<0.001
Prostatectomy specimen ISUP Gleason Grading						
0	0 (0%)	1 (0.5%)	<0.001	0 (0%)	0 (0%)	<0.001
1	11 (5.0%)	13 (5.9%)		4 (3.2%)	1 (0.8%)	
2	68 (30.8%)	163 (73.8%)		15 (12.1%)	51 (41.1%)	
3	75 (33.9%)	38 (17.2%)		28 (22.6%)	41 (33.1%)	
4	21 (9.5%)	3 (1.4%)		34 (27.4%)	22 (17.7%)	
5	46 (20.8%)	3 (1.4%)		43 (34.7%)	9 (7.3%)	
pT stage						
T0	0 (0%)	1 (0.3%)	0.930	0 (0%)	0 (0%)	0.254
T2	184 (80.6%)	185 (84.2%)		87 (70.2%)	92 (74.2%)	
T3a	29 (15.2%)	29 (12.9%)		30 (24.2%)	21 (16.9%)	
T3b	8 (3.8%)	6 (2.5%)		7 (5.6%)	11 (8.9%)	
pN stage						
N0	0 (0%)	213 (96.4%)		0 (0%)	115 (92.7%)	-
N1	0 (0%)	8 (3.6%)		0 (0%)	9 (7.3%)	
Nx	221 (100%)	0 (0%)		124 (100%)	0 (0%)	
Surgical margin						
positive	34 (15.4%)	20(9.0%)	0.00246	24 (19.4%)	23 (18.5%)	0.325
unknown	2 (0.9%)	13(5.9%)		0 (0%)	3 (2.4%)	

**Table 3 cancers-17-00655-t003:** Comparison of the Clavien–Dindo classification grade of perioperative complications in the original cohort.

Complications	ePLND (–) (*n* = 461)	ePLND (*n* = 541)	*p* Value
All Grade	37 (8.0%)	159 (29.4%)	
1	12 (2.6%)	89 (16.5%)	<0.0001
2	15 (3.3%)	37 (6.8%)	
3	10 (2.2%)	33 (6.1%)	
<3	27	126	
3≤	10	33	
Complications related with PLND	0 (0%)	69 (12.8%)	
extremity edema	0 (0%)	53 (9.8%)	<0.001
Pelvic hematoma	0 (0%)	9 (1.7%)	0.00467
Neuropathy	0 (0%)	7 (1.3%)	0.0174

**Table 4 cancers-17-00655-t004:** Univariate and multivariable analyses for biochemical recurrence for the patients at intermediate or high risk, a propensity-score-matched cohort.

	Intermediate Risk			High Risk		
Variable	Univariate *p* Value	Multivariable		Univariate *p* Value	Multivariable	
		HR 95% CI	*p* Value		HR 95% CI	*p* Value
Age (years)	0.32			0.85		
PSA (ng/mL)						
<10	ref			ref		
10–20	0.018	0.38 (0.18–0.81)	0.013	<0.001	1.97 (1.00–3.86)	0.050
20<	n.a.	n.a.	n.a.	0.014	2.87 (1.07–7.70)	0.037
ePLND yes	0.62	1.31 (0.55–3.07)	0.54	0.95	1.06 (0.54–2.07)	0.87
Prostatectomy specimen ISUP Gleason Grading						

0	ref			ref		
1	1.0			0.88		
2	0.99			0.14		
3	0.99			0.90		
4	0.99			n.a		
5	0.99			n.a		
pT stage						
T0	ref					
T2	0.99			ref	ref	Ref
T3a	0.99			<0.001	2.22 (1.08–4.56)	0.029
T3b	0.99			0.0011	1.98 (0.77–5.08)	0.155
Surgical margin						
negative	ref	ref	ref	ref	ref	ref
positive	0.002	3.68 (1.67–8.10)	0.001	<0.001	2.41 (1.24–4.70)	0.010
unknown	0.072	3.68 (0.82–16.5)	0.088	0.99	n.a	n.a.

n.a.: not applicable.

## Data Availability

The raw data supporting the conclusions of this article will be made available on request.

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
