# Peer review of "Extended Lymph Node Dissection May Not Provide a Therapeutic Benefit in Patients with Intermediate-to High-Risk Prostate Cancer Treated with Robotic-Assisted Radical Prostatectomy"

_cancers, 2025, doi:10.3390/cancers17040655_

Round 1
Reviewer 1 Report
Comments and Suggestions for Authors
Dear authors,
As suggested, I reviewed the paper entitled “Extended Lymph Node Dissection May Not Provide a Therapeutic Benefit in Patients with Intermediate- to High-risk Prostate Cancer Treated with Robotic-assisted Radical Prostatectomy”. Although the topic addressed is of major importance and the references are accurate, the paper shows several meaningful limitations that might reduce its potential value for readers.
As regards the statistical analysis in page 3 line 106 Authors stated that “After creating a pseudo-population, background imbalances between groups were removed using a propensity score-based method to estimate the mean treatment effect without bias.” In this light, a more granular definition of the propensity scoring method should be provided (for example “nearest neighbor”) as well as the matching ration (for example 1:2). Moreover, all the matching features should be better described and the propensity matching represented in a dedicated plot.
Regarding the intraoperative features, the authors should better highlight the presence of a non-negligible bias in terms of learning curve between the two assessed centers. In this light, the median difference of >100 minutes in the console time is probably not only related with the LND template given the different times reported in literature for this phase. Similarly, the presence of a significantly worse pathological stage might be driven by the presence of significative differences in the preoperative staging assessment between the two centers.
In terms of postoperative complications, a more extensive description of the follow up assessment should be provided. Particularly, the “neuropathy” occurrence should be better specified.
Regarding the perioperative outcomes after minimally invasive radical prostatectomy, might be of clinical interest mentioning the following paper (DOI: 10.1016/j.ejso.2024.108741)
Comments on the Quality of English Language
Extensive revision is required
Author Response
We thank you and the reviewers for your thoughtful suggestions and insights. The manuscript has benefited from these insightful suggestions. I look forward to working with you and the reviewers to move this manuscript closer to publication in Cancers.
The manuscript has been rechecked, and the necessary changes have been made in accordance with the reviewers’ suggestions. The responses to all comments have been prepared and given below.
Thank you for your consideration. I look forward to hearing from you.
Responses to the Comments by the reviewer 1:
Reviewer1
As suggested, I reviewed the paper entitled “Extended Lymph Node Dissection May Not Provide a Therapeutic Benefit in Patients with Intermediate- to High-risk Prostate Cancer Treated with Robotic-assisted Radical Prostatectomy”. Although the topic addressed is of major importance and the references are accurate, the paper shows several meaningful limitations that might reduce its potential value for readers.
1)As regards the statistical analysis in page 3 line 106 Authors stated that “After creating a pseudo-population, background imbalances between groups were removed using a propensity score-based method to estimate the mean treatment effect without bias.” In this light, a more granular definition of the propensity scoring method should be provided (for example “nearest neighbor”) as well as the matching ration (for example 1:2). Moreover, all the matching features should be better described and the propensity matching represented in a dedicated plot.
REPLY: Thank you for your important suggestion.
We performed this analysis using the Nearest Neighbor Matching method with a 1:1 matching ratio. The caliper was set to 0.2 standard deviations of the propensity score.
We added these details to the methods session. (Page3 Line 114-116)
In addition, we showed the propensity matching expressed in a dedicated plot at a figure1. (Page4 Line 141-142)
2) Regarding the intraoperative features, the authors should better highlight the presence of a non-negligible bias in terms of learning curve between the two assessed centers. In this light, the median difference of >100 minutes in the console time is probably not only related with the LND template given the different times reported in literature for this phase. Similarly, the presence of a significantly worse pathological stage might be driven by the presence of significative differences in the preoperative staging assessment between the two centers.
REPLY: Thank you for your important suggestion. With regard to the duration of surgery, at St Luke's International Hospital, a peritoneal suturing and anterior reinforcement were performed, on the other hand these were not performed at Ehime University. These factors may have contributed to the prolonged operation time. However, we think that these factors do not affect oncological outcomes.
In terms of the quality of lymph node dissection, St. Luke's International Hospital is one of the facilities that performs high-quality dissection for ePLND for PCa in Japan.
According to reports from various high-volume centers, the median number of removed lymph nodes was 18 in Morizane S et al. (IJCO 27, 781-89, 2022), 18 in Abdollah F et al. (BJU-I 108, 1769-75, 2011). And 20 nodes were removed in ePLND for NCCN very high risk PCa at Ehime University (Cancer medicine 22, 7968-76, 2021).
Similarly, the median number of removed lymph nodes was 18 at st Luke's International Hospital. Therefore, we believe that the number of removed lymph nodes is almost the same, and there are no technical differences.
Regarding to clinical staging, every patients recruited our study has the clinical evaluation by MRI. The median number of biopsies was 13 in the PLND(-) group and 15 in the PLND(+) group, and as the number of biopsies was slightly lower in the PLND(-) group, it is possible that the frequency of up-grading of the GS of the prostatectomy specimens was higher in the PLND(-) group. However, despite the ePLND(-) group having worse factors, there was no significant difference between two groups.
3) In terms of postoperative complications, a more extensive description of the follow up assessment should be provided. Particularly, the “neuropathy” occurrence should be better specified.
REPLY: Thank you for your important suggestion. As reviewer has pointed out, I add the details of the follow-up regarding postoperative complications as follows.
(Page 5 Line 170~173 in results section)
Of the 53 cases of lower limb edema, 6 cases (11.3%) required the long-term use of stock-ings. Intra-pelvic hematoma was observed in eight cases. Seven cases improved with con-servative treatment, and one case underwent drainage. Peripheral sensory neuropathy was observed in seven cases, but all cases improved spontaneously.
4) Regarding the perioperative outcomes after minimally invasive radical prostatectomy, might be of clinical interest mentioning the following paper (DOI: 10.1016/j.ejso.2024.108741)
REPLY: Thank you for your recommendation. We mention the report by Lambertin which showed perioperative outcomes of frail prostate cancer patients, then we added as follows.
Page 7 Line 240-241 Discussion session.
Furthermore, it has been reported that the frequency of complications in frail patients exceeds 50% [27], and in cases where the risk of complications due to surgical invasion or the risk of disadvantages due to prolonged surgery time are high, omitting lymph node dissection might be considered.
Comments on the Quality of English Language
Extensive revision is required
REPLY: Thank you for your suggestion. I’m sorry I didn’t express it clearly in English.
I have requested English editing services from MDPI Author Services.
Reviewer 2 Report
Comments and Suggestions for Authors
The authors evaluated clinical significance of ePLND of Pca patients who received RARP using two high volume centers in Japan by PSM. Basically the article is well-written and understandable but there needs to be some revisions before acceptance.
l What percentage of patients clinically diagnosed using MRI before surgery?
l Although the authors used PSM to evaluate the significance of ePLND using pre-operative factors, there were significant differences in pathological ISUP Gleason grade grouping between ePLND (+) and ePLND (-) both in intermediate-risk and high-risk group. According to the distribution of pathological ISUP Gleason grade grouping between ePLND (+) and ePLND (-), patients with ePLND (-) might be higher in bPFS if they had lower pathological ISUP Gleason grade grouping. The authors should also evaluate PSM using pathological factors such as pathological ISUP Gleason grade grouping, pT stage and RM. Since pathological factors are more significant for bPFS than clinical factors.
l Biological PFS strongly depends on surgical techniques of RARP. How many expert surgeons performed ePLND (-) and ePLND (+)? Are there any differences between them?
l Follow-up period was too short to definitively compare the bPFS between ePLND (+) and ePLND (-) since most cases with censoring existed between follow-up period 0 months to 20 months. So it is too strong to definitively concluded that ePLND does not affect survival (bPFS). The authors should revise the description of their conclusions.
Minor
l Please check that there is a discrepancy in description about median numbers of lymph node dissection between L 136 and L179.
l The Y axis of Figure 1 b lacks “probability of 1.0”. Did the authors intentionally delete it? Why?
Author Response
We thank you and the reviewers for your thoughtful suggestions and insights. The manuscript has benefited from these insightful suggestions. I look forward to working with you and the reviewers to move this manuscript closer to publication in Cancers.
The manuscript has been rechecked, and the necessary changes have been made in accordance with the reviewers’ suggestions. The responses to all comments have been prepared and given below.
Responses to the Comments by the reviewer 2
Comments and Suggestions for Authors
The authors evaluated clinical significance of ePLND of Pca patients who received RARP using two high volume centers in Japan by PSM. Basically, the article is well-written and understandable but there needs to be some revisions before acceptance.
l) What percentage of patients clinically diagnosed using MRI before surgery?
REPLY: Thank you for your suggestion. All patients in the study were diagnosed by MRI before surgery. We added this sentence in the results section.(Page 3, Line 125-126)
2) Although the authors used PSM to evaluate the significance of ePLND using pre-operative factors, there were significant differences in pathological ISUP Gleason grade grouping between ePLND (+) and ePLND (-) both in intermediate-risk and high-risk group. According to the distribution of pathological ISUP Gleason grade grouping between ePLND (+) and ePLND (-), patients with ePLND (-) might be higher in bPFS if they had lower pathological ISUP Gleason grade grouping. The authors should also evaluate PSM using pathological factors such as pathological ISUP Gleason grade grouping, pT stage and RM. Since pathological factors are more significant for bPFS than clinical factors.
REPLY: "Surgical margin" has the same meaning as "resection margin (RM)". Since they were mixed in the text, we unified them to surgical margin(Table 2). I apologize for the confusing expression.
As shown in the results, in the intermediate-risk group, ISUP GS and surgical margin were significantly worse in the ePLND(-) than in ePLND(+). Similarly, in the high-risk group, ISUP GS was significantly worse in the ePLND(-) group than in ePLND(+) group.
Thus, despite the ePLND(-) group having worse factors, there was no significant difference between two groups.
Furthermore, a trend-adjusted multivariate COX regression analysis including prostatectomy specimen ISUP Gleason grading, pT stage, and SM was performed to eliminate the effect of confounding factors as much as possible.
Therefore, I consider it highly unlikely that these confounding factors influenced the result.
3) Biological PFS strongly depends on surgical techniques of RARP. How many expert surgeons performed ePLND (-) and ePLND (+)? Are there any differences between them?
REPLY: At St Luke's International Hospital, three expert surgeons performed RARP with ePLND as operators or supervisors, and at Ehime University Hospital, six expert surgeons performed RARP without ePLND as operators or supervisors.
At both facilities, the specialist surgeons managed the surgeries; there was no difference in the number of cases between the two facilities and no difference in the skills of the specialist surgeons at the two facilities.
4) Follow-up period was too short to definitively compare the bPFS between ePLND (+) and ePLND (-) since most cases with censoring existed between follow-up period 0 months to 20 months. So it is too strong to definitively concluded that ePLND does not affect survival (bPFS). The authors should revise the description of their conclusions.
REPLY:As noted by the reviewers, we revised the conclusions as follows.
Abstract session (Page2, Line 45-51)
The oncological outcomes of patients did not differ significantly between patients with IR or HR according to the National Comprehensive Cancer Network (NCCN) risk classification, regardless of whether ePLND was performed during RARP. However, ePLND increased the surgical time and associated complications.
→
These results suggest that ePLND may not be necessary in intermediate- to high-risk PCa patients undergoing RARP, although further study with long follow-up is required.
Main session (Page8, Line 261- 266)
The oncological outcomes of patients did not differ significantly between patients with IR or HR according to the NCCN risk classification, regardless of whether ePLND was performed during RARP. In contrast, performing ePLND increased the surgical time and risk of complications. Thus, future randomized controlled trials evaluating the therapeutic significance of ePLND to clarify patient selection and the extent of dissection are warranted.
→
These results suggest that ePLND may not be necessary in intermediate- to high-risk PCa patients undergoing RARP, although further study with long follow-up is required. Thus, future randomized controlled trials evaluating the therapeutic significance of ePLND to clarify patient selection and the extent of dissection are warranted.
Minor
l ) Please check that there is a discrepancy in description about median numbers of lymph node dissection between L 136 and L179.
REPLY: Thank you for your suggestion. “18” is correct number of lymph node dissection.
I revise that. (Page 5, Line 154 results section)
In the ePLND group, a median of 19 18 nodes was dissected (interquartile range: 14–25).
2) The Y axis of Figure 1 b lacks “probability of 1.0”. Did the authors intentionally delete it? Why?
REPLY: Thank you for your suggestion. This is a simple mistake. I add “1.0”.
(Page 6 Line 190-193)
Figure 2. Kaplan–Meier curves for BCR-free survival with and without lymph node dissection in the propensity-score-matched cohort of intermediate- (a) and high-risk (b) patients, according to the NCCN.
Round 2
Reviewer 2 Report
Comments and Suggestions for Authors
The authors responded to my requests appropriately.